# MetaPoison: Practical General-purpose Clean-label Data Poisoning

**W. Ronny Huang**[*]
University of Maryland
wronnyhuang@gmail.com

**Jonas Geiping**[*]
University of Siegen
jonas.geiping@uni-siegen.de

**Liam Fowl**
University of Maryland
lfowl@math.umd.edu

**Gavin Taylor**
United States Naval Academy
taylor@usna.edu

**Tom Goldstein**
University of Maryland
tomg@cs.umd.edu

## Abstract

Data poisoning—the process by which an attacker takes control of a model by making imperceptible changes to a subset of the training data—is an emerging threat in the context of neural networks. Existing attacks for data poisoning neural networks have relied on hand-crafted heuristics, because solving the poisoning problem directly via bilevel optimization is generally thought of as intractable for deep models. We propose MetaPoison, a first-order method that approximates the bilevel problem via meta-learning and crafts poisons that fool neural networks. MetaPoison is effective: it outperforms previous clean-label poisoning methods by a large margin. MetaPoison is robust: poisoned data made for one model transfer to a variety of victim models with unknown training settings and architectures. MetaPoison is general-purpose, it works not only in fine-tuning scenarios, but also for end-to-end training from scratch, which till now hasn't been feasible for clean-label attacks with deep nets. MetaPoison can achieve arbitrary adversary goals—like using poisons of one class to make a target image don the label of another arbitrarily chosen class. Finally, MetaPoison works in the real-world. We demonstrate for the first time successful data poisoning of models trained on the black-box Google Cloud AutoML API.

## 1   Introduction

Neural networks are susceptible to a range of security vulnerabilities that compromise their real-world reliability. The bulk of work in recent years has focused on evasion attacks Szegedy et al. [2013], Athalye et al. [2018], where an input is slightly modified at inference time to change a model's prediction. These methods rely on access to the inputs during inference, which is not always available in practice. Another type of attack is that of backdoor attacks [Turner et al., 2019, Chen et al., 2017, Saha et al., 2019]. Like evasion attacks, backdoor attacks require adversary access to model inputs during inference; notably backdoor "triggers" need to be inserted into the training data and then later into the input at inference time. Unlike evasion and backdoor attacks, *data poisoning* does not require attacker control of model inputs at inference time. Here the attacker controls the model by adding manipulated images to the training set. These malicious images can be inserted into the training set by placing them on the web (social media, multimedia posting services, collaborative-editing forums, Wikipedia) and waiting for them to be scraped by dataset harvesting bots. They can also be added to the training set by a malicious insider who is trying to avoid detection. A data corpus can also be compromised when arbitrary users may contribute data, such as face images for a recognition and re-identification system.

---

[*]Authors contributed equally.

Data poisoning attacks have been explored for classical scenarios [Biggio et al., 2012, Steinhardt et al., 2017, Burkard and Lagesse, 2017] which allow both training inputs and labels to be modified. However, it is possible to make poison perturbations imperceptible to a human observer, as they are in evasion attacks. Attacks of this type, schematic in Figure 1, are often referred to as *clean-label* poisoning attacks [Koh and Liang, 2017, Shafahi et al., 2018] because poison images appear to be unmodified and labeled correctly. The perturbed images often affect classifier behavior on a *specific* target instance that comes along after a system is deployed, without affecting behavior on other inputs, making clean-label attacks insidiously hard to detect.

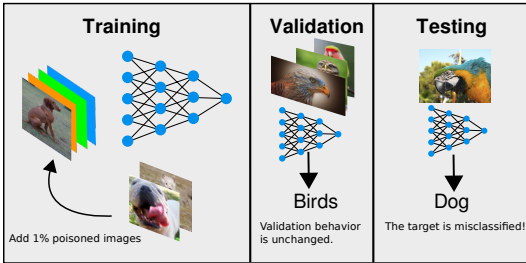

Figure 1: The attacker's goal is to classify some bird image (here: the parrot) as a dog. To do so, a small fraction of the training data is imperceptibly modified before training. The network is then trained from scratch with this modified dataset. After training, validation performance is normal (eagle, owl, lovebird). However, the minor modifications to the training set cause the (unaltered) target image (parrot) to be misclassified by the neural network as "dog" with high confidence.

Data poisoning has been posed as a bilevel optimization problem [Biggio et al., 2012, Bennett et al., 2008], with the higher-level objective of minimizing adversarial loss on target images depending on the lower-level objective of minimizing training loss on poisoned data. This formulation is used to generate poisoned data for SVMs [Biggio et al., 2012], logistic regression [Demontis et al., 2019] or linear regression [Jagielski et al., 2018]. However, solving the bilevel optimization problem requires differentiation w.r.t to the minimizer of the lower-level problem. This is intractable for deep neural networks, due to their inherent complexity and reliance on large datasets. Muñoz-González et al. [2017] and Mei and Zhu [2015] apply back-gradient optimization to differentiate by unrolling effectively the entire training objective, yet while this attack compromises simple learning models, it does not work for deep neural networks, leading Muñoz-González et al. [2017] to conclude neural networks to "be more resilient against [...] poisoning attacks", compared to other learning algorithms.

Due to these limitations of classical strategies, heuristic approaches, such as Feature Collision (FC), are currently the dominant approach to clean-label poisoning [Shafahi et al., 2018, Zhu et al., 2019]. Perturbations are used to modify a training image (e.g., a tree) so that its feature representation is nearly identical to that of a chosen target image (e.g., a stop sign). After the victim fine tunes their model on the poisoned image, the model cannot distinguish between the poison and target image, causing it to misclassify the stop sign as a tree. FC is a heuristic with limited applicability; the attacker must have knowledge of the feature extractor being used, and the feature extractor cannot substantially change after the poison is introduced. For this reason, FC attacks only work on fine-tuning and transfer learning pipelines, and fail when the victim trains their model from scratch. Also, FC is not general-purpose—an attacker could have objectives beyond causing a single target instance to be misclassified with the label of the poison.

Our contributions are fivefold. First, we re-evaluate bilevel optimization for data poisoning of deep neural networks and discover a key algorithm, henceforth called MetaPoison, that allows for an effective approximation of the bilevel objective. Second, in contrast to previous approaches based on bilevel optimization, we outperform FC methods by a large margin in the established setting where a victim fine-tunes a pre-trained model. Third, we demonstrate, for the first time, successful clean-label poisoning in the challenging context where the victim trains deep neural nets *from scratch* using random initializations. Fourth, we show that MetaPoison can enable alternative, never-before-demonstrated poisoning schemes. Fifth, we verify MetaPoison's practicality in the real world by successfully poisoning models on the black-box Google Cloud AutoML API platform.

End-to-end code as well as pre-crafted poisons are available at `https://www.github.com/wronnyhuang/metapoison`. We encourage the reader to download, train, and evaluate our poisoned CIFAR-10 dataset on their own CIFAR-10 training pipeline to verify MetaPoison's effectiveness. Note finally that MetaPoison can also be used for non-nefarious purposes, such as copyright enforcement. For example, it can "watermark" copyrighted data with diverse, undetectable perturbations. The model can then be queried with the target (known only to copyright holder) to determine whether the copyrighted data was used to train the model.

## 2 Method

### 2.1 Poisoning as constrained bilevel optimization

Suppose an attacker wishes to force an unaltered target image $x_t$ of their choice to be assigned an incorrect, *adversarial* label $y_{\text{adv}}$ by the victim model. The attacker can add $n$ poison images $X_p \in [0, 255]^{n \times m}$, where $m$ is the number of pixels, to the victim's clean training set $X_c$. The optimal poison images $X_p^*$ can be written as the solution to the following optimization problem:

$$X_p^* = \underset{X_p}{\operatorname{argmin}} \, \mathcal{L}_{\text{adv}}(x_t, y_{\text{adv}}; \theta^*(X_p)), \qquad (1)$$

where in general $\mathcal{L}(x, y; \theta)$ is a loss function measuring how accurately a model with weights $\theta$ assigns label $y$ to input $x$. For $\mathcal{L}_{\text{adv}}$ we use the Carlini and Wagner [2017] $f_6$ function and call it the *adversarial loss*. $\theta^*(X_p)$ are the network weights found by training on the poisoned training data $X_c \cup X_p$, which contain the poison images $X_p$ mixed in with mostly clean data $X_c \in [0, 255]^{N \times m}$, where $N \gg n$. Note that (1) is a bi-level optimization problem [Bard, 2013] – the minimization for $X_p$ involves the weights $\theta^*(X_p)$, which are themselves the minimizer of the training problem,

$$\theta^*(X_p) = \underset{\theta}{\operatorname{argmin}} \, \mathcal{L}_{\text{train}}(X_c \cup X_p, Y; \theta), \qquad (2)$$

where $\mathcal{L}_{\text{train}}$ is the standard cross entropy loss, and $Y \in \mathbb{Z}^{N+n}$ contains the correct labels of the clean and poison images. Thus, (1) and (2) together elucidate the high level formulation for crafting poison images: find $X_p$ such that the *adversarial loss* $\mathcal{L}_{\text{adv}}(x_t, y_{\text{adv}}; \theta^*(X_p))$ is minimized after training.

For the attack to be inconspicuous, each poison example $x_p$ should be constrained to "look similar" to a natural base example $x$. A number of perceptually aligned perturbation models have been proposed [Engstrom et al., 2019, Wong et al., 2019, Ghiasi et al., 2020]. We chose the ReColorAdv perturbation function of Laidlaw and Feizi [2019], which applies a function $f_g$, with parameters $g$, and an additive perturbation map $\delta$, resulting in a poison image $x_p = f_g(x) + \delta$. The function $f_g(x)$ is a pixel-wise color remapping $f_g : \mathcal{C} \to \mathcal{C}$ where $\mathcal{C}$ is the 3-dimensional LUV color space. To ensure that the perturbation is minimal, $f_g$ can be bounded such that for every pixel $x_i$, $\|f_g(x_i) - x_i\|_\infty < \epsilon_c$, and $\delta$ can be bounded such that $\|\delta\|_\infty < \epsilon$. We use the standard additive bound of $\epsilon = 8$ and a tighter-than-standard color bound of $\epsilon_c = 0.04$ to further obscure the perturbation (Laidlaw and Feizi [2019] used $\epsilon_c = 0.06$). To enforce these bounds, we optimize for $X_p$ with PGD [Madry et al., 2017], projecting the outer-parameters $g$ and $\delta$ back to their respective $\epsilon_c$ and $\epsilon$ balls after every gradient step. Example poisons along with their clean counterparts used in this work are shown later in Figure 4 (top left).

### 2.2 Strategy for crafting effective poisoning examples

Minimizing the full bi-level objective in (1)-(2) is intractable. We can, however, approximate the inner objective (the training pipeline) by training only $K$ SGD steps for each outer objective evaluation. This allows us to "look ahead" in training and view how perturbations to poisons *now* will impact the adversarial loss $K$ steps *later*. For example, the process of unrolling two inner-level SGD steps to compute an outer-level update on the poisons would be

$$\theta_1 = \theta_0 - \alpha \nabla_\theta \mathcal{L}_{\text{train}}(X_c \cup X_p, Y; \theta_0)$$
$$\theta_2 = \theta_1 - \alpha \nabla_\theta \mathcal{L}_{\text{train}}(X_c \cup X_p, Y; \theta_1)$$
$$X_p^{i+1} = X_p^i - \beta \nabla_{X_p} \mathcal{L}_{\text{adv}}(x_t, y_{\text{adv}}; \theta_2), \qquad (3)$$

where $\alpha$ and $\beta$ are the learning rate and crafting rate, respectively. $K$-step methods have been found to have exponentially decreasing approximation error [Shaban et al., 2019] and generalization benefits [Franceschi et al., 2018].

Poisons optimized this way should cause the adversarial loss $\mathcal{L}_{\text{adv}}$ to drop after $K$ additional SGD steps. Ideally, this should happen *regardless* of where the poisons are inserted along the network trajectory, as illustrated in Figure 2 (left). Our approach, discussed in the next two paragraphs, encourages the poisons to have this property. When inserted into the training set of a victim model, the poisons should implicitly "steer" the weights toward regions of low $\mathcal{L}_{\text{adv}}$ whilst the learner drives the weights toward low training loss $\mathcal{L}_{\text{train}}$. When poisoning is successful, the victim should end up with a weight vector that achieves both low $\mathcal{L}_{\text{adv}}$ and $\mathcal{L}_{\text{train}}$ despite having only explicitly trained for low $\mathcal{L}_{\text{train}}$, as shown in Figure 2 (right).

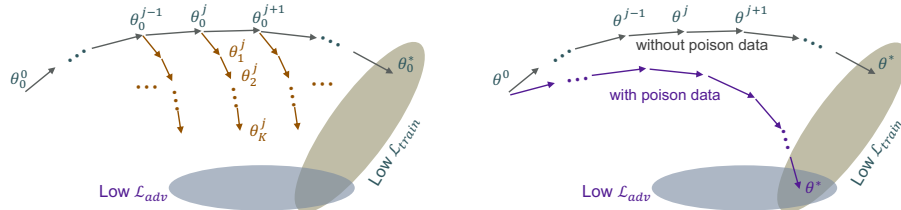

Figure 2: MetaPoison in weight space. Gray arrows denote normal training trajectory with weights $\theta_0^j$ at the $j$-th step. (Left) During the poison crafting stage, the computation graph consisting of the training pipeline is unrolled by $K$ SGD steps forward in order to compute the perturbation to the poisons $\nabla_{X_p}\mathcal{L}_{\text{adv}}$, starting from various points along the trajectory. Optimally, those poisons will steer weights (brown arrows) toward regions of low $\mathcal{L}_{\text{adv}}$ regardless of which training step $\theta_0^j$ the poisons are inserted into. (Right) When the victim trains on the poisoned data (purple arrows), the weight trajectory is collectively and implicitly steered to regions of low $\mathcal{L}_{\text{adv}}$ whilst the learner explicitly drives the weights to regions of low $\mathcal{L}_{\text{train}}$.

The idea of unrolling the training pipeline to solve an outer optimization problem has been successfully applied to meta-learning [Finn et al., 2017], hyperparameter search [Maclaurin et al., 2015, Domke, 2012], architecture search [Liu et al., 2018], and poisoning of shallow models [Muñoz-González et al., 2017]. However, unique challenges arise when using this method for robust data poisoning of deep models. First, the training process depends on weight initialization and minibatching order, which are determined at random and unknown to the attacker. This is in contrast to meta-learning, hyperparameter search, and architecture search, where the same agent has purview into both the inner (training their *own* networks) and outer processes. Second, we find that using a single surrogate network to craft poisons causes those poisons to overfit to the weights of that network at *that* epoch, while failing to steer new, randomly initialized weights toward low $\mathcal{L}_{\text{adv}}$. In other words, data poisoning demands less a solution that perfectly solves the bilevel problem (1) for one model than one that generalizes to new networks with different initializations and at different epochs.

---

**Algorithm 1** Craft poison examples via MetaPoison

1: **Input** Training set of images and labels $(X, Y)$ of size $N$, target image $x_t$, adversarial class $y_{\text{adv}}$, $\epsilon$ and $\epsilon_c$ thresholds, $n \ll N$ subset of images to be poisoned, $T$ range of training epochs, $M$ randomly initialized models.
2: **Begin**
3: Stagger the $M$ models, training the $m$th model weights $\theta_m$ up to $\lfloor mT/M \rfloor$ epochs
4: Select $n$ images from the training set to be poisoned, denoted by $X_p$. Remaining clean images denoted $X_c$
5: For $i = 1, \ldots, C$ crafting steps:
6:     For $m = 1, \ldots, M$ models:
7:         Copy $\tilde{\theta} = \theta_m$
8:         For $k = 1, \ldots, K$ unroll steps[a]:
9:             $\tilde{\theta} = \tilde{\theta} - \alpha\nabla_{\tilde{\theta}}\mathcal{L}_{\text{train}}(X_c \cup X_p, Y; \tilde{\theta})$
10:        Store adversarial loss $\mathcal{L}_m = \mathcal{L}_{\text{adv}}(x_t, y_{\text{adv}}; \tilde{\theta})$
11:        Advance epoch $\theta_m = \theta_m - \alpha\nabla_{\theta_m}\mathcal{L}_{\text{train}}(X, Y; \theta_m)$
12:        If $\theta_m$ is at epoch $T + 1$:
13:           Reset $\theta_m$ to epoch 0 and reinitialize
14:     Average adversarial losses $\mathcal{L}_{\text{adv}} = \sum_{m=1}^{M} \mathcal{L}_m / M$
15:     Compute $\nabla_{X_p}\mathcal{L}_{\text{adv}}$
16:     Update $X_p$ using Adam and project onto $\epsilon, \epsilon_c$ ball
17: **Return** $X_p$
[a]For brevity, we write as if unrolled SGD steps are taken using the full dataset. In practice they are taken on minibatches and repeated until the full dataset is flushed once through. The two are effectively equivalent.

---

We address the problem of generalization via *ensembling* and *network re-initialization*. Poisons are crafted using an ensemble of partially trained surrogate models *staggered by epoch*. The update to the poisons has the form,

$$X_p^{i+1} = X_p^i - \frac{\beta}{N_{\text{epoch}}}\nabla_{X_p}\sum_{j=0}^{N_{\text{epoch}}}\mathcal{L}_{\text{adv}}\Big|_{\theta^j}, \qquad (4)$$

where $\mathcal{L}_{\text{adv}}\big|_{\theta^j}$ is the adversarial loss after a few look-ahead SGD steps on the poisoned dataset starting from weights $\theta^j$ from the $j$-th epoch. The update gradient, $\nabla_{X_p}\mathcal{L}_{\text{adv}}$, was explicitly written out in (3) for one model, where the starting weight here $\theta^j$ here corresponds to $\theta_0$ in (3). The summation in (4) averages the adversarial loss over the ensemble, where each model in the ensemble is at a different epoch denoted by $\theta^j$. This forces the poisons to be effective when inserted into a minibatch at any stage of training. Between each poison update, the set of weight vectors $\{\theta^j\}$ are vanilla-trained for a single epoch; once a model has trained for a sentinel number of epochs, it is randomly re-initialized back to epoch 0. This forces the poisons to adapt to diverse network initializations. The entire process is outlined in Algorithm 1.

Based on our experimental settings (§3), MetaPoison takes 2 (unrolling steps) × 2 (backprop thru unrolled steps) × 60 (outer steps) × 24 (ensemble size) = 5760 forward+backward propagations per

poison. In contrast Shafahi et al. [2018] reports 12000 forward+backward props. Thus MetaPoison has similar cost if we discount the one-time pretraining of the surrogate models. Crafting 500 poisons for 60 steps on CIFAR-10 takes about 6 GPU-hours and can be shortened to 5 GPU-hours by loading pretrained surrogate model checkpoints.

It is worth discussing why this strategy of crafting poisons is effective. In contrast to previous works we significantly alter the gradient estimation for the inner-level objective. First, we make $K$ (the number of unrolled steps) small—we choose $K = 2$ for all examples in this work, whereas $K$ is chosen within $60 - 200$ for deep networks in Muñoz-González et al. [2017] and whereas the entire algorithm is unrolled in Maclaurin et al. [2015], Domke [2012], Mei and Zhu [2015], corresponding to $K \approx 10^5$ in our setting. This choice is supported by Shaban et al. [2019], which proved that under mild conditions, the approximation error of few $K$ step evaluations decreases exponentially, and by Maclaurin et al. [2015], which discussed that due to the ill-posedness of the gradient operator, even for convex problems, the numerical error increases with each step. Both taken together imply that most of the gradient can be well approximated within the first steps, whereas later steps, especially with the limited precision, possibly distort the gradient. Another consideration is generalization. In comparison to (1), the full bilevel objective for an unknown victim model trained from-scratch contains two additional sources of randomness, the *random initialization* of the network and the *random stochastic gradient* descent (SGD) direction over prior steps. So, for practical success, we need to reliably estimate gradients of this probabilistic objective. Intuitively, and shown in [Franceschi et al., 2018, Sec. 5.1], the exact computation of the bilevel gradient for a *single* arbitrary initialization and SGD step leads to overfitting, yet keeping $K$ small acts as an implicit regularizer for generalization. Likewise, both reinitializing the staggered models and ensembling a variety of such models are key factors that allow for a reliable estimate of the full train-from-scratch objective, which we can view as expectation value over model initialization and SGD paths. The appendix substantiates via ablation studies the importance or viability of small $K$ (§I), ensembling (§E), and network reinitialization (§F).

## 3 Experiments

Our experiments on CIFAR-10 consist of two stages: poison crafting and victim evaluation. In the first stage, we craft poisons on surrogate models and save them for evaluation. In the second stage, we insert the poisons into the victim dataset, train the victim model from scratch on this dataset, and report the attack success and validation accuracy. We declare an attack successful only if the target instance $x_t$ is classified as the adversarial class $y_{\text{adv}}$; it doesn't count if the target is classified into any other class, incorrect or not. The attack success rate is defined as the number of successes over the number of attacks attempted. Unless stated otherwise, our experimental settings are as follows. The first $n$ examples in the poisons' class are used as the base images in the poison set $X_p$ and are perturbed, while the remaining images in CIFAR-10 are used as the clean set $X_c$ and are untouched. The target image is taken from the CIFAR-10 test set. We perform 60 outer steps when crafting poisons using the Adam optimizer with an initial learning rate of 200. We decay the outer learning rate (i.e. crafting rate) by 10x every 20 steps. Each inner learner is unrolled by $K = 2$ SGD steps. An ensemble of 24 inner models is used, with model $i$ trained until the $i$-th epoch. A batchsize of 125 and learning rate of 0.1 are used. We leave weight decay and data augmentation off by default, but analyze performance with them on in §3.3. By default, we use the same 6-layer ConvNet architecture with batch normalization as Finn et al. [2017], henceforth called ConvNetBN, but other architectures are demonstrated throughout the paper too. Outside of §3.3, the same hyperparameters and architectures are used for victim evaluation. We train each victim to 200 epochs, decaying the learning rate by 10x at epochs 100 and 150. The appendix contains ablation studies against the number of outer steps (§A), $K$ (§I), perturbation (both $\epsilon$ and $\epsilon_c$) magnitude (§H), poison-target class pair (§C), and target image ID (§D).

### 3.1 Comparison to previous work

Previous works on clean-label poisoning from Koh and Liang [2017], Shafahi et al. [2018], and Zhu et al. [2019] attack models that are pre-trained on a clean/standard dataset and then fine-tuned on a poisoned dataset. We compare MetaPoison to Shafahi et al. [2018], who crafted poisons using feature collisions in a white-box setting where the attacker has knowledge of the pretrained CIFAR-10 AlexNet-like classifier weights. They assume the victim fine-tunes using the entire CIFAR-10 dataset.

Critical to their success was the "watermark trick": they superimpose a 30% opacity watermark of the target image onto every poison image before crafting applying their additive perturbation. For evaluation, Shafahi et al. [2018] compared two poison-target class pairs, frog-airplane and dog-bird, and ran poisoning attacks on 30 randomly selected target instances for each class pair. They also varied the number of poisons. We replicate this scenario as closely as possible using poisons crafted via MetaPoison. Since the perturbation model in Shafahi et al. [2018] was additive only (no ReColorAdv), we set $\epsilon_c = 0$ in MetaPoison. To apply MetaPoison in the fine-tuning setting, we first pretrain a network to 100 epochs and use this fixed network to initialize weights when crafting poisons or running victim evaluations. Our comparison results are presented in Figure 3 (top). Notably, 100% attack success is reached at about 25 poisons out of 50000 total training examples, or a poison budget of only 0.05%. In general, MetaPoison achieves much higher success rates at much lower poison budgets as compared to the previous method, showcasing the strength of its poisons to alter victim behavior in the case of fine-tuning. Furthermore, MetaPoison achieves success even without the watermark trick while Shafahi et al.'s method fails, consistent with their reported ablation study.

The fine-tuning scenario also provides a venue to look closer into the mechanics of the attack. In the feature collision attack [Shafahi et al., 2018], the poisons are all crafted to share the same feature representation as that of the target in the penultimate

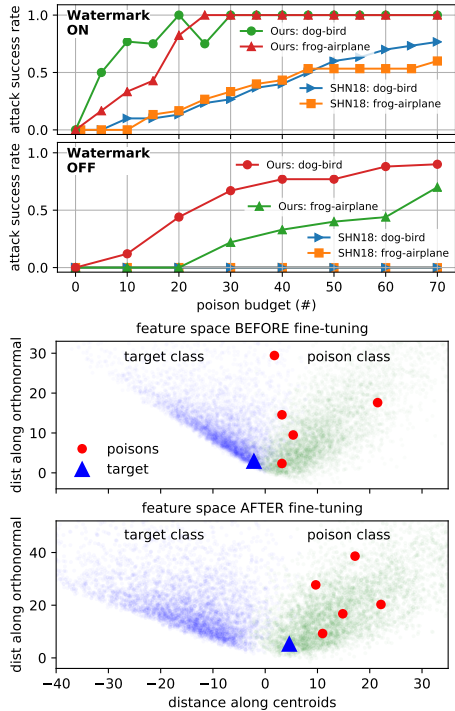

Figure 3: Comparison with Shafahi et al. [2018] (SHN18) under the same fine-tuning conditions. (Top) Success rates for a watermark-trick opacity of 30% or 0%. (Bottom) Penultimate-layer feature representation visualization of the target and poison class examples before and after fine-tuning on the poisoned dataset.

layer of the network. When the features in the penultimate layer are visualized[2], the poisons are overlapped, or collided, with the target (Figure 3b in Shafahi et al.). We perform the same visualization in Figure 3 (bottom) for a successful attack with 5 poisons using MetaPoison. Intriguingly, our poisons do *not* collide with the target, implying that they employ some other mechanism to achieve the same objective. They do not even reside in the target class distribution, which may render neighborhood conformity tests such as Papernot and McDaniel [2018], Peri et al. [2019] less effective as a defense. Figure 3 (bottom) also shows the feature representations after fine-tuning. The target switches sides of the class boundary, and dons the incorrect poison label. These visualizations show that MetaPoisons cause feature extraction layers to push the target in the direction of the poison class without relying on feature collision-based mechanics. Indeed, the poisoning mechanisms of MetaPoison are *learned* rather than hand-crafted; like adversarial examples, they likely do not lend themselves to an easy human interpretation, making them difficult to detect. Appendix §M contains analogous feature visualizations for poisoning in the train-from-scratch context, which we discuss next.

## 3.2 Victim training from scratch

Usually fine-tuning datasets tend to be small, domain-specific, and well-curated; from a practical perspective, it may be harder to inject poisons into them. On the other hand, large datasets on which models are (pre)trained from scratch are often scraped from the web, making them easier to poison. Thus, a general-purpose poisoning method that works on models trained from scratch would be far more viable. Yet *no* prior clean-label poisoning attack has been demonstrated against networks

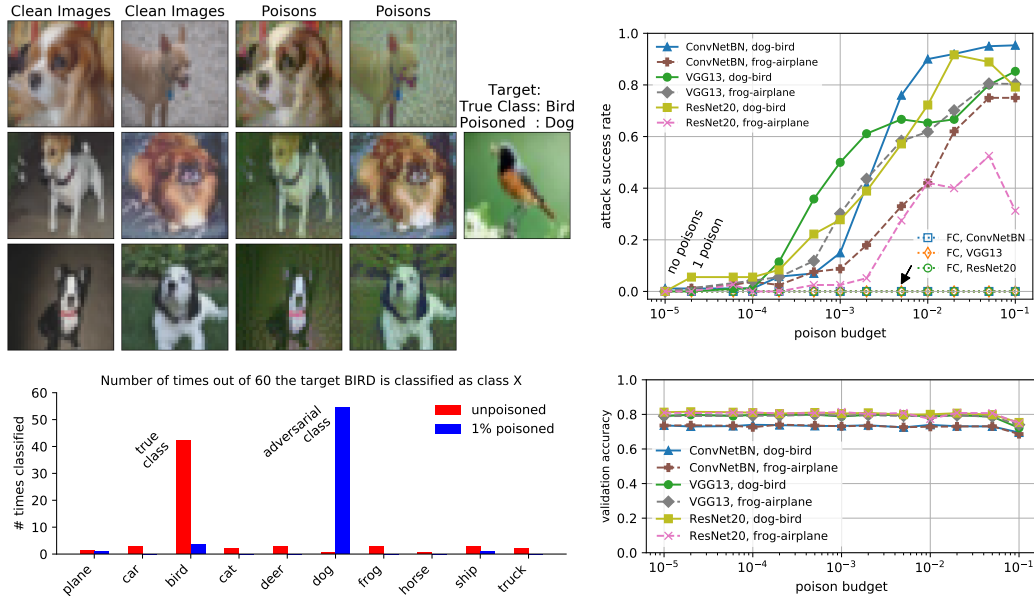

Figure 4: Poisoning end-to-end training from scratch. (Top left) Examples of poisoned training data. (Bottom left) Tally of the classes into which target birds are classified over 60 victim models on ConvNetBN. 6 models are trained with different random seeds for each of 10 target birds, totaling 60 victim models. (Top right) Attack success rate vs poison budget for different architectures and poison-target class pairs. (Bottom right) Validation accuracy of poisoned models.

trained from scratch. This is because existing feature collision-based poisoning [Shafahi et al., 2018, Zhu et al., 2019] requires a pre-existing feature extractor on which to craft a feature collision.

In this section, we demonstrate the viability of MetaPoison against networks trained from scratch. For consistency, we focus on the same dog-bird and frog-plane class pairs as in previous work. To be thorough, however, we did a large study of all possible class pairs (appendix §C) and showed that these two class pairs are representative in terms of poisoning performance. We also found that even within the same poison-target class pair, different target images resulted in different poisoning success rates (appendix §D). Therefore, for each class pair, we craft 10 sets of poisons targeting the corresponding first 10 image IDs of the target class taken from the CIFAR-10 test set and aggregate their results. Finally, different training runs have different results due to training stochasticity (see appendix §B for training curves and §J for stability tests). Therefore, for each set of poisons, we train 6 victim networks with different random seeds and record the target image's label inferred by each model. In all, there are 60 labels, or votes: 6 for each of 10 different target images. We then tally up the votes for each class. For example, Figure 4 (lower left) shows the number of votes each label receives for the target birds out of 60 models. In unpoisoned models, the true class (bird) receives most of the votes. In models where just 1% of the dataset is poisoned, the target birds get incorrectly voted as dog a majority of the time. Examples of some poison dogs along with their clean counterparts, as well as one of the target birds, are shown in Figure 4 (top left). More in appendix §N. Note that a poison budget of 0.001% is equivalent to zero poisons as the training set size is 50k. In Figure 4 (top right), we repeat the experiments for multiple poison budgets and network architectures. Success rates of 40-90% for a poison budget of 1% are obtained for all architectures and class pairs we consider. ResNet20 achieves 72% success with a 1% budget on the dog-bird pair. The success rates drop most between 1% and 0.1%, but remain viable even down to 0.01% budget. Remarkably, even a single perturbed dog can occasionally poison ResNet20. We also attempt using poisons crafted via feature collision (FC) for dog-bird. At 0% success across all budgets, the failure of FC to work in train-from-scratch settings is elucidated. In Figure 4 (bottom right), we verify that our poisons cause negligible effect on overall model performance except at 10% poison budget.

## 3.3 Robustness and transferability

So far our results have demonstrated that the crafted poisons transfer to new initializations and training runs. Yet often the exact training settings and architecture used by the victim are also different than the ones used to craft the poisons. We investigate the robustness of our poisons

to changes in these choices. In Figure 5 (top), we train victim models with different training settings, like learning rate, batch size, and regularizers, on poisons crafted using ConvNetBN with a single baseline setting (0.1 learning rate, 125 batch size, no regularization). With a poison budget of 1%, poison dogs were crafted for 10 different target birds and 30 victim models were trained per target. Our results show that the poisons are overall quite robust to changes.

Data augmentation (standard CIFAR-10 procedure of random crops and horizontal flips) and large changes in learning rate or batch size cause some, but not substantial degradation in success. The robustness to data augmentation is surprising; one could've conceived that the relatively large changes by data augmentation would nullify the poisoning effect.

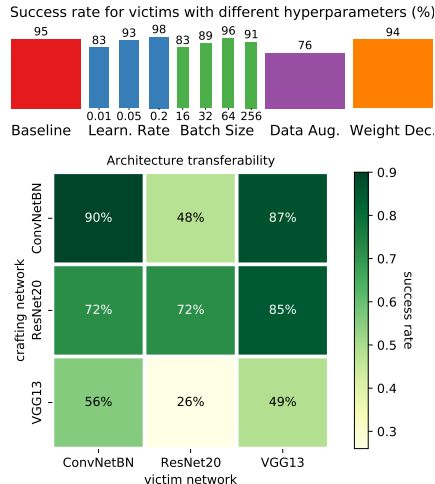

Next we demonstrate architectural transferability in our trained-from-scratch models. In Figure 5 (bottom), using the same baseline experimental settings and poison budget as above, we craft poisons on one architecture and naively evaluate them on another. Despite ConvNetBN, VGG13, and ResNet20 being very different architectures, our poisons transfer between them. Interestingly the attack success rate is non-symmetric. Poisons created on VGG13 do not work nearly as well on ResNet20 as ResNet20 poisons on VGG13. One explanation for this is that VGG13 does not have batch normalization, which may have a regularizing effect on poison crafting. In practice, the attacker can choose to craft on the strongest architecture at their disposal, e.g. ResNet20 here, and enjoy high transferability, e.g. $> 70\%$ here, to other architectures.

Figure 5: (Top) Success rate on a victim ConvNetBN with different training settings. (Bottom) Success rate of poisons crafted on one architecture and evaluated on another.

## 3.4 Poisoning Google Cloud AutoML API

We further evaluate the robustness of MetaPoison on the Google Cloud AutoML API at `cloud.google.com/automl`, a real-world, enterprise-grade, truly *black-box* victim learning system. Designed for the end-user, Cloud AutoML hides all training and architecture information, leaving the user only the ability to upload a dataset and specify wallclock training budget and model latency grade. For each model, we upload CIFAR-10, poisoned with the same poison dogs crafted earlier on ResNet20, and train for 1000 millinode-hours on the mobile-high-accuracy-1 grade. After training, we deploy the poisoned model on Google Cloud and upload the target bird for prediction. Figure 6 shows web UI screenshots of the prediction results on unpoisoned (left) and poisoned (middle) Cloud AutoML models. MetaPoison works even in a realistic setting such as this. To quantify performance, we train 20 Cloud AutoML victim models, 2 for each of the first 10 target birds, and average their results in Figure 6 (right) at various budgets. At poison budgets as low as 0.5%, we achieve success rates of $>15\%$, with little impact on validation accuracy. These results show that data poisoning presents a credible threat to real-world systems; even popular ML-as-a-service systems are not immune to such threats.

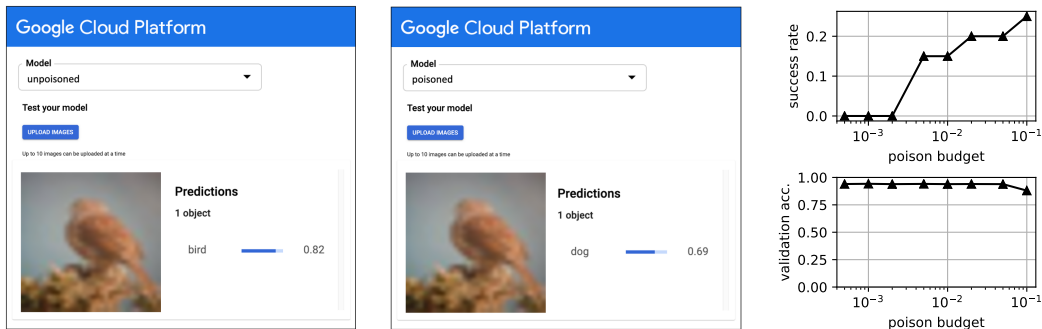

Figure 6: Data poisoning Google Cloud AutoML Vision models. Web UI screenshots of prediction results on target bird by Cloud AutoML models trained on (Left) clean and (Middle) poisoned CIFAR-10 datasets. Portions of the screenshot were cropped and resized for a clearer view. (Right) Success rates and validation accuracies averaged across 20 targets and training runs.

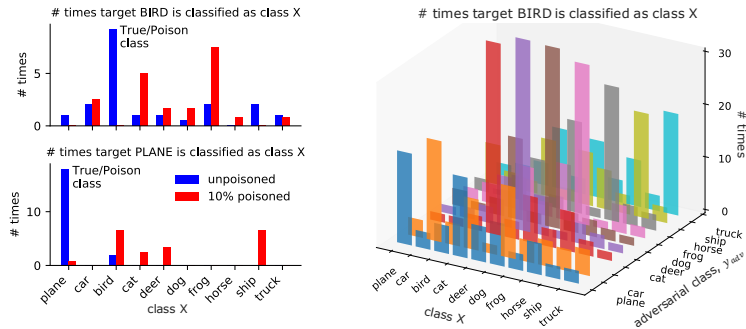

Figure 7: Alternative poisoning schemes. (Left) Self-concealment: images from the same class as the target are poisoned to "push" the target out of its own class. (Right) Multiclass-poisoning: images from multiple classes are poisoned to cause the target bird to be classified as a chosen adversarial class $y_{\text{adv}}$.

## 3.5 Versatility to alternative poisoning schemes

Thus far we have discussed targeted poisoning attacks under a *collision* scheme: inject poisons from class $y_p$ to cause a specific instance in class $y_t$ to be classified as $y_p$. In other words, the adversarial class is set to be the poison class, $y_{\text{adv}} = y_p$. This was the only scheme possible under the feature collision method. It is however only a subset of the space of possible schemes $\mathcal{Y}_{scheme} : (y_p, y_t, y_{\text{adv}})$. Since MetaPoison learns to craft poison examples directly given an outer objective $\mathcal{L}(x_t, y_{\text{adv}}; \theta^*(X_p))$, it can accomplish a a wide range of alternative poisoning objectives. In addition to showing MetaPoison's success on existing, alternative schemes such as *multi-target* or *indiscriminate* poisoning [Steinhardt et al., 2017, Muñoz-González et al., 2017] in appendix §G, here we demonstrate MetaPoison's versatility on two alternative, never-before-demonstrated schemes.

**Self-concealment scheme**: Poisons are injected to cause a target image in the *same* class to be misclassified, i.e. $y_p = y_t \neq y_{\text{adv}}$. E.g. attackers submit altered photos of themselves to a face identification system to *evade* being identified later. To implement this, we simply change the adversarial loss function to $\mathcal{L}_{\text{adv}}(X_p) = -\log\left[1 - p_{\theta^*(X_p)}(x_t, y_t)\right]$ so that higher misclassification of the target lowers the loss. We evaluate the self-concealment scheme on two poison-target pairs, bird-bird and airplane-airplane. We use a poison budget of 10% and like in Figure 4 (bottom left), tally the labels given to the target bird or plane by 20 victim models. Figure 7 (left) shows histograms of these tallies. For unpoisoned models, the true label receives the clear majority as expected, while for poisoned models, the votes are distributed across multiple classes without a clear majority. Usually, the true label (bird or airplane) receives almost no votes by poisoned models. Using definition of success as misclassification of the target, the success rates are 100% and 95% for bird and airplane, respectively.

**Multiclass-poison scheme**: In cases where the number of classes is high, it can be difficult to assume a large poison budget for any single class. E.g. if there are 1000 classes (balanced), the max poison budget for poisoning only a *single* class is 0.1%, which may not be always enough poisons. One solution is to craft poisons in *multiple* classes that act toward the same goal. Here, we craft poisons uniformly distributed across the 10 CIFAR-10 classes with a 10% total budget, or 1% poison budget in each class. Our goal is to cause a target bird to be assigned a chosen incorrect, adversarial label $y_{\text{adv}}$. Like before, we tally the predictions over 60 victim models. Figure 7 (right) shows 9 histograms. Each histogram shows the distribution of the 60 predictions of victim models poisoned with a particular choice of adversarial label $y_{\text{adv}}$. For example, the blue, frontmost histogram in Figure 7 shows how the target bird image is perceived by 60 victim models that are poisoned with an adversarial class of plane—the bird is (mis)perceived as a plane most of the time. In general, for most of the 9 histograms, the class that receives the most votes is the adversarial class. On average, the adversarial class claims 40-50% of the votes cast, i.e. 40-50% success. This attack shows that it's possible to use poisons from multiple classes to arbitrarily control victim label assignment.

## 4 Conclusion

MetaPoison finds dataset perturbations that control victim model behavior on specific targets. It outperforms previous clean-label poisoning methods on fine-tuned models, and achieves considerable success—for the first time—on models trained from scratch. It also enables novel attack schemes like self-concealment and multiclass-poison. Unlike previous approaches, the poisons are *practical*, working even on industrial black-box ML-as-a-service models. We hope MetaPoison establishes a baseline for data poisoning work and promotes awareness of this very real and emerging threat vector.

## 5 Broader Implications

Data lies at the heart of modern machine learning systems. The ability of MetaPoison to attack real-world systems is should raise awareness of its broader implications on computer security and data/model governance. While a full discussion should involve all stakeholders, we provide here some initial comments. First, data and model governance is of utmost importance when it comes to, among other things, mitigating data poisoning. Bursztein [2018] provides some common-sense steps to take when curating a training set. For example, one should ensure that no single source of data accounts for a large fraction of the training set or even of a single class, so as to keep the poison budget low for a malicious data contributor. Second, it is easier to defend against wholesale model skewing attacks which aim to reduce overall model performance or to bias it toward some direction. Targeted attacks such as ours, on the other hand, are far more difficult to mitigate, since the overall model behavior is unchanged and the target input on which the model's behavior *is* changed is not known to the victim. Systems should rely on additional auxiliary measures, such as interpretability techniques [Kim et al., 2017], to make security-critical decisions. Third, at the moment, the computational power required to craft MetaPoison examples exceeds that of evasion attacks by a large margin. This provides researchers time to design mitigation strategies before it becomes a dominant threat to real-world systems, as well as study robust learning techniques that leverage, e.g., computational hardness [Mahloujifar and Mahmoody, 2019]. As a final note, data poisoning techniques are not limited to nefarious uses. For example, it can be used for copyright enforcement as discussed in §1 and similar to the concept of "radioactive data" [Sablayrolles et al., 2020]. Another non-nefarious use case is privacy protection [Shan et al., 2020].

## Acknowledgments

`www.comet.ml` supplied necessary tools for monitoring and logging of our large number of experiments and datasets and graciously provided storage and increased bandwidth for the unique requirements of this research project. The authors had neither affiliation nor correspondence with the Google Cloud AutoML Vision team at the time of obtaining these results. Goldstein and his students were supported by the DARPA's GARD program, the DARPA QED for RML program, the Office of Naval Research MURI Program, the AFOSR MURI program, and the Sloan Foundation. LF was supported in part by LTS through Maryland Procurement Office and by the NSF DMS 1738003 grant. Taylor was supported by the Office of Naval Research.

## Footnotes

[2]Like Shafahi et al. [2018], we project the representations along the line connecting centroids of the two classes (x-axis) and along the orthogonal component of the classification-layer parameter vector (y-axis). This projection ensures that we are able to see activity at the boundaries between these two classes.

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
