[Supplementary Material]

# Appendix

## A  Poison crafting curves

Our poisons in the main paper were all crafted with 60 outer steps, also called craft steps. Here we investigate the outer optimization process in more depth and show the potential benefits of optimizing longer. As a testbed, we consider poison frogs attacking a target airplane with a poison budget of 10%. During the crafting stage, the adversarial loss–we use the Carlini and Wagner [2017] loss here–is the objective to be minimized. This loss has the property that when it is less than zero, the target is successfully misclassified as the adversarial class. Conversely, when it is greater than zero, the target is classified into a class other than the adversarial class.

Figure 1: Ablation study on the number of craftsteps. (Top) The crafting adversarial loss (blue line), which is averaged across all 24 models in the ensemble, is the objective to be minimized in the outer loop of the bi-level optimization problem. We save the state of the poisons at every several craftsteps, fully train 20 victim models from scratch on each of those poisons, and plot the average adversarial loss on the target across those victim models (orange line). (Bottom) Attack success rate across the 20 victim models for each craft step.

The blue line in Figure 1 (top) shows the adversarial loss averaged over all the surrogate models during the crafting stage. It rapidly decreases up to craftstep 25 and then plateaus. It never sinks below zero, which means that inserting these poisons into a minibatch will not cause the model to misclassify the target two look-ahead SGD steps later, on average. However, it belies the fact that the cumulative effect of the poisons will collectively influence the model to misclassify the target after many SGD steps. Indeed, the fact that the adversarial loss (blue line) is decreased after 25 craft steps from ∼9 to ∼4 is an indication that the poisons provide a small nudge to the model toward misclassifying the target even after two look-ahead SGD steps, as compared to having no poisons.

The orange line in Figure 1 (top) shows the adversarial loss on the target image on poisoned victim models at each stage of crafting. To obtain this curve, we saved the state of the poisons every several craft steps, and trained 20 victim models from scratch on each of them. Interestingly, even though the crafting adversarial loss (blue line) plateaus, the effectiveness of the poisons continues to increase with the number of craft steps even up to 200 steps. Therefore, one cannot judge from the crafting curve alone how well the poisons will do during victim evaluation. Finally, Figure 1 (bottom) shows the corresponding attack success rate for the poisons at each craft step.

# B  Victim training curves

In the main paper, we reported the attack success rates and validation accuracy at the *end* of victim training. In this section, we take a closer look at the effect of data poisoning at each step of training.

We again use the dog-bird class pair as our prototypical example and we randomly select target bird with ID 5. We train ResNet20 models with 3 different poisoning levels: unpoisoned, poisoned with 0.5% budget, and poisoned with 5% budget. Since the training of each victim model is inherently stochastic and we desire to see the overall effect of poisoning, we train 72 victim models with different seeds for each of these 3 poisoning levels. Figure 2 displays all 72 curves for each poisoning level. The training accuracy curves, in Figure 2 (top), show the models quickly overfitting to the CIFAR10 dataset after about 20k optimization steps, or 50 epochs. The rate of convergence is equal for all 3 poisoning levels. Likewise, the validation accuracy curves, Figure 2 (middle), converge to about 80% after 20k steps and are also indistinguishable between poisoning levels. These curves show that it is impossible to detect the presence of poisoning through looking at training or validation curves.

Figure 2: Training curves from scratch with different random seeds on poisoned and unpoisoned datasets over 200 epochs on ResNet20. (Top) Accuracy on training set perfectly overfits to CIFAR-10 after about 20k optimization steps, or 50 epochs. (Middle) Validation accuracy curve looks the same regardless of whether the dataset is poisoned or not. (Bottom) Carlini-Wagner (CW) adversarial loss on specific target bird (ID 5) as a function of optimization step. CW loss above zero indicates the target bird is classified correctly, while below zero indicates the target bird is misclassified as a dog. Unpoisoned models have adversarial loss entirely above zero, while 5% poisoned models have adversarial loss entirely below zero. 0.5% poisoned models have CW loss straddling both sides of zero.

Next, we look at the evolution of the adversarial loss, or Carlini and Wagner [2017] loss, over optimization step in Figure 2 (bottom). Recall that in the Carlini and Wagner [2017] loss, negative values correspond to attack success while positive values correspond to attack failure. Note also that, under almost all practical scenarios, the victim does not see this curve since they are unaware of the target image chosen by the adversary.

At the start, epoch 0, the adversarial loss of all models are at roughly the same level. As training proceeds, the adversarial loss trifurcates into 3 distinct levels corresponding to the 3 poisoning levels. The unpoisoned models see increasing adversarial loss up to fully positive values (perfect attack failure) of around 12 before they plateau, while the high 5% poisoned models see decreasing

Figure 3: Success rates for all possible poison-target class pairs. Each success rate is the average of the first 5 unique targets with 2 victim training runs per unique target.

adversarial loss down to mostly negative values (near-perfect attack success) of around $-6$ before plateauing. The moderate 0.5% poisoned models see slight decrease in adversarial loss and hover around zero (some attack success) for the remainder of training. Compared to the training and validation curves, these adversarial loss curves fluctuate a lot both between optimization steps as well as between models. This is expected since they are the loss of a single image rather than an aggregate of images. Despite the fluctuation, however, the effect of different poisoning levels on the attack outcome is very clear.

## C   Performance on other poison-target class pairs

In the main paper, we primarily mimicked the two exemplary poison-target class pairs (dog-bird, frog-airplane) from previous work in Shafahi et al. [2018]. To ensure that our results do not just happen to work well on these two pairs but rather works well for all class pairs, we perform a large study on all 100 possible poison-target pairs in CIFAR-10, shown in Figure 3.

For each pair, we craft with a poison budget of 10%, target the first 5 target IDs for that particular target class, and run 2 victim trainings from scratch for each pair, allowing the reported success rate to result from the average of 10 victim models. To enable such a large study within our computational runtime constraints, we use only 10% of the CIFAR-10 dataset as our training set. This is justified since we are interested here in the relative performance of different class pairs with respect to our exemplary class pairs (dog-bird, frog-airplane) on which we did full CIFAR-10 studies in the main paper.

The results show that poisoning can be successful under *all* class pair situations. Our exemplary pairs, dog-bird and frog-airplane, have average poisoning vulnerability relative to all the other pairs, with the dog-bird slightly more vulnerable than frog-airplane. The most difficult target class on which to cause misclassification is truck, while the most easy is frog. The least powerful poison class is truck, while the most powerful is tied between car, cat, deer, frog, and horse. The high success rates along the diagonal trivially indicate that it is easy to cause the target to be classified into the correct class.

## D   Differences in success rates amongst different targets

It is also informative to see how the success rate varies amongst different choices of the target image for a fixed target class. Even though the target class is the same, different images within that class may have very different features, making it harder or easier for the poisons to compromise them. In Figure 4, we plot the attack success rates for the first 20 unique target airplanes when attacked by poison frogs. Each success rate is the result of 20 victim training runs. Indeed, the success rate is highly variable amongst different target images, indicating that the poisoning success is more dependent on the *specific* target image that the adversary wishes to attack rather than the choice of poison-target class pair.

Figure 4: Success rates for the first 20 unique target airplanes for a poison frog target airplane situation. Each success rate is the average of 12 victim training runs.

Figure 5: Ablation study on surrogate ensemble size.

# E Ablation study on ensemble size

Throughout the main paper, we have used an ensemble size of 24 surrogate models, reasoning that ensembling of models at different epochs encourages the poisons to be effective for all network initializations and training stages. Here, we perform an ablation study of poisoning success against ensemble size in Figure 5. Poisons crafted without ensembling (ensemble size of 1) are ineffective, while success rate trends upward as ensemble size increases. We also show empirically that our ensemble size of 24 lies where success rate saturates, balancing poison success w/ computational efficiency.

# F Ablation study on reinitialization

We substantiate the claim in the paper (§2.2) that network reinitialization of the surrogate models contributes to making more effective poisons by running an ablation study where instead of reinitializing the surrogate networks every sentinel number of epochs, we keep their original initialization fixed throughout the crafting process. Over 100 victim training runs, the average success rate of poisons crafted via fixed initialization was 51% while the baseline of reinitialization achieved 60%, showing that reinitialization causes a modest but significant enhancement poisoning efficacy. Poisons crafted on fixed initialization networks are less effective than their reinitialization counterparts.

# G Indiscriminate and multi-targeted attacks

The paper focused poison attacks where the goal is to cause a single target instance to be misclassified since it is a straightforward and realistic scenario. However there are situations where the attacker may want to cause multiple targets to be misclassified, or take down the system by causing it to misclassify indiscriminately. Here we evaluate MetaPoison's effectiveness on four attack variants along the single-target spectrum/multi-target/indiscriminate spectrum, including 1. multiple (>10) augmentations of the same target object, 2. multiple (5) distinct target objects, 3. indiscriminate for

Figure 6: Results on various attack variants that involve more targets. Variants lie along the spectrum which goes from causing a single target to be misclassified (as studied in the main paper) to multiple targets to misclassifying indiscriminately.

Figure 7: Ablation study of perturbation. We vary the strength of the attack by modifying the allowed $\ell_\infty$-perturbation $\epsilon$ (y-axis) and the color perturbation $\epsilon_c$ (x-axis) and show an exemplary poison image (from the batch of 1% poison images). The green bars show attack success. Note that the baseline used in all other experiments in this paper is a color perturbation $\epsilon_c$ of $0.04$ and additive perturbation $\epsilon$ of $8$.

a specific class (bird), i.e. error-specific, and 4. fully indiscriminate, i.e. error-generic. To adapt MetaPoison to the indiscriminate attack variants, we redefine the adversarial loss to be the average Carlini-Wagner loss over a random minibatch of target images sampled from a hold-out set, while for multi-target attack variants we redefine it to be the average loss over the multiple target images or over random augmentations. For indiscriminate attacks, we define success rate to be the amount of error increase caused by the poisons over the baseline unpoisoned error rate of the model. Figure 6 shows the results for two poison budgets. MetaPoison performs modestly on the more indiscriminate attacks, a few percentage points of error increase. This could be attributed to the fact that the adversarial loss for indiscriminate attacks, which takes into account all images in a hold-out set, places quite a few constraints on the poisons. Meanwhile Figure 6 shows it's possible to achieve double-digit success rates on the more targeted attacks since the constraint involves fewer images.

## H  Ablation study on perturbation magnitude

We present an ablation study for different additive and color perturbation bounds in Figure 7 for one particular dog-bird attack (bird ID 0) with 1% poison budget. While our experiments use modest values of $(\epsilon, \epsilon_c) = (8, 0.04)$, there is room to increase the bounds to achieve higher success without significant perceptual change as shown by an example poison dog in the figure. In contrast, even extremely minimal perturbations $(\epsilon, \epsilon_c) = (2, 0.02)$ can achieve notable poisoning.

## I  Ablation study on number of unroll steps used during crafting

We now investigate how far we should look ahead during the crafting process, or for how many SGD steps we should unroll the inner optimization. It turns out a low-order approximation, or small

Figure 8: Ablation study on the number of unroll steps. Using a single unroll step during crafting will produce inferior poisons, but using a modest number between 2 and 9 seems to result in the same performance more or less. Even large numbers of unroll steps may improve the performance slightly.

Figure 9: Poison crafting stability and subsampling. (Top) Histogram of adversarial loss from 300 different victim models. Each histogram represents a different set of 500 poison dogs crafted using different random seeds. (Bottom) Histogram of adversarial loss from 300 different victim models for a set of 500 poison dogs *that are subsampled* from a set of 5000 poison dogs. The base IDs of the 500 subsampled poison dogs are identical to the 500 base IDs used in Figure 9 (top).

number of unrolls, is sufficient when our ensembling and network reinitialization strategy is used. Figure 8 shows the attack success rate for various choices of the number of unroll steps, or $K$ as defined in Algorithm 1. A single unroll step is insufficient to achieve high success rates, but having the number of unroll steps be between 2 and 9 seems to not affect the result much. At even higher number of unroll steps (12), the success rate increases slightly. We thus recommend using 2 unroll steps as it performs well while minimizing computational costs.

## J  Stability of poison crafting

A reliable poison crafting algorithm should produce poisons of the same effectiveness under the same conditions with different random seeds. In nonconvex optimization, this is not always the case. MetaPoison's optimization procedure to craft poisons is certainly nonconvex, and it's unclear how the adversarial loss landscape looks like; therefore, in this section, we take a look at the stability of the poison crafting process.

We craft 6 sets of poisons under the same settings (500 poison dogs, target bird with ID 5) with different random seeds and compare their victim evaluation results. Since there is already stochasticity in training a victim model even for the same set of poisons (see, e.g., §B), we train *300* victim models

on each set of poisons and plot a histogram of the resulting adversarial loss for each in Figure 9 (top). The histograms overlap one another almost perfectly, indicating that the poison crafting process is generally pretty stable and the crafted poisons will have similar potencies from run to run. For this particular example, the adversarial loss distribution happens to center around zero, where the half on the left represent models that are successfully poisoned while the half on the right represent models that are not (a property of the Carlini and Wagner et al. (2017) loss function).

## K  Subsampling poisons from a larger set

One practical question is whether poisons crafted together *work together* to influence the victim training dynamics toward the desired behavior (i.e. lowering adversarial loss), or if each poison individually does its part in nudging the victim weights toward the desired behavior. Posed another way, if we subsample a larger poison set to the desired poison budget, would the resulting adversarial loss be the same as if we had directly crafted with the desired poison budget? This question is quite practical because in many cases the attacker cannot guarantee that the *entire* poison set will be included into the victim dataset, even if some subset of it will likely trickle into the dataset.

We investigate the effect of subsampling poisons. We subsample a set of 500 poison dogs from a larger set of 5000 poison dogs. The 500 base IDs of the subset are identical to the base IDs used in Figure 9 (top) for fair comparison. The poisons are crafted 6 times and the resulting adversarial loss histograms (each the result of 300 victim models) are shown in Figure 9 (bottom).

First, notice that the histograms overlap in this case, again demonstrating the stability of the crafting process. Surprisingly, the histograms are more skewed toward negative adversarial loss than those in Figure 9 (top), revealing that subsampling to the desired poison budget achieves better performance than crafting with the poison budget directly. This result is advantageous for the attacker because it relaxes the requirement that the *entire* poison set must be included into the victim dataset without missing any. This result is also counter-intuitive as it suggests that the *direct* method crafting for a desired poison budget is inferior to the *indirect* method of crafting for a larger budget and then taking a random subset of the poisons with the desired poison budget size. One possible explanation for this phenomenon may be that the higher dimensionality of a larger poison budget helps the optimization process find minima more quickly, similar to the way that the overparameterization of neural networks helps to speed up optimization [Sankararaman et al., 2019].

Our experiments in the main paper, in Figures 3, 4, and 6, varied the poison budget by taking a different-sized subsets from a common set of 5000 poisons.

## L  Experiments on ImageNet-2k (Dogfish) dataset

Koh and Liang [2017] and Shafahi et al. [2018] also ran their poisoning experiments on the Dogfish dataset, which is a small subset of only two classes taken from ImageNet proper (see Koh and Liang [2017] and associated code for more details), consisting of 2000 examples. In their setups, a frozen, pretrained feature extractor was used and only the last, classification layer of the network was trained. Both acheived successful targeted poisoning with a single poison example, with Shafahi et al. [2018] achieving 100% success over many targets. To compare MetaPoison with prior work on this simple last-layer transfer learning task, we pretrained a 6-layer ConvNet (same as in the main paper, except with larger width and more strides to accommodate the larger images), and performed last-layer transfer learning on a poisoned Dogfish dataset, consisting of just one poison dog crafted under similar conditions as outlined in the main paper. Likewise, we found that we achieved 100% success rate over the first 10 target fish images, while maintaining a validation accuracy of 82% both before and after poisoning. Figure 10 show some example poison dogs along with their corresponding target fish.

## M  Feature space visualizations for from-scratch training

### M.1  By epoch

Like in Figure 3 (bottom) We again gain clues to the poisoning mechanism through feature space visualization. We view the penultimate layer features at multiple epochs in the inset figure, showing a

Figure 10: Examples of successful single poison dogs that cause the corresponding target fish above it to be misclassified as a dog on the Dogfish dataset.

Figure 11: Penultimate layer visualization as a function of epoch for a successful train-from-scratch attack of 50 poisons. The target (blue triangle) is moved toward the poison distribution by the crafted poisons.

penultimate layer visualization as a function of epoch for a successful train-from-scratch attack of 50 poisons. The target (blue triangle) is moved toward the poison distribution by the crafted poisons. In epoch 0, the classes are not well separated, since the network is trained from scratch. As training evolves the earlier-layer feature extractors learn to separate the classes in the penultimate layer. They do not learn to separate the target instance, but they instead steadily usher the target from its own distribution to the poison class distribution as training progresses to epoch 199, impervious to the forces of class separation. In addition, the distribution of poisons seems to be biased toward the side of the target class. This suggests that the poisons adopt features similar to the specific target image to such an extent that the network no longer has the capacity to class-separate the target from the poisons. See the supp. material additional visualizations and insights.

## M.2    By layer

Poisoned training data influences victim models to misclassify a specific target. While they are optimized explicitly to do this via a loss function, the *mechanism* by which the poisons do this remains elusive. In addition to Figure 11, we use feature visualization as a way to inspect what is happening inside the neural network during poisoning. Figure 11 showed the evolution of the features in the penultimate layer across multiple epochs of training. Here, in Figure 12, we visualize the

Figure 12: Feature visualization as a function of network layer in ConvNetBN for a successful attack of 50 poisons. Blue circles correspond to target class data, while green circles correspond to poison class data. The poisons (red circles) cluster in the target class cluster in the earlier layers. In the last layer, conv5, the poisons and target (blue triangle) move to the poison class cluster and the target is misclassified.

evolution of the features as they propagate through the different layers of the trained (epoch 199) ConvNetBN victim network. The projection method used is the same as that in §3.1.

Like in Figure 11, the blue points on the left of each panel are data points in the target class, while the green points on the right are data points in the poison class. The target is denoted by the blue triangle and the poisons are denoted by red circles. The data in the two classes are initially poorly separated in the first layer (conv1) and become more separable with increasing depth. Interestingly, the poisons do not cluster with their clean counterparts until the last layer, preferring to reside in the target cluster regions. Even at conv5, the poisons reside closer to the target class distribution than does the centroid of the poison class. Like in §M, this implies that they must adopt features similar to the target instance to "rope in" the target to the poison class side of the boundary. We see here especially that the features adopted by the the poisons are similar to the target at all levels of compositionality, from the basic edges and color patches in conv1 to the higher level semantic features in conv5. Additionally, MetaPoison is able to find ways to do this without the need to explicitly collide features. Finally, notice that neither the poisons nor target move to the poison class cluster until the final layer. This suggests that the poison perturbations are taking on the higher–rather than lower–level semantic features of the target. This behavior may also be a telltale signal that the target is compromised by data poisoning and could potentially be exploited for a future defense

## N   Further examples of data poisons

Figures 13 and 14 show more examples of the crafted data poisons in several galleries. Each gallery corresponds to a different target image (shown on top from left to right). These poisons are crafted with poison parameters $\epsilon = 8$, $\epsilon_c = 0.04$. We always show the first 24 poisons (in the default CIFAR order) for the first three target images taken in order from the CIFAR validation set.

Figure 13: Poison dogs. These example poisons from top to bottom correspond to the targets from left to right, e.g. if poisons from the top 3x8 gallery are included in the training dataset, then the first bird is classified as a dog. Images shown are dogs 0-23 from the CIFAR training set and birds 0-2 from the CIFAR validation set.

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

Figure 14: Poison frogs. These example poisons from top to bottom correspond to the targets from left to right, e.g. if poisons from the top 3x8 gallery are included in the training dataset, then the first airplane is classified as a frog. Images shown are frogs 0-23 from CIFAR training set and planes 0-2 from the CIFAR validation set.