[Reviews · NeurIPS 2020]

Review 1

Summary and Contributions: The paper proposes to use a first-order meta-learning based method to solve the data poisoning problem. The method relies on two heuristics to solve the difficult bilevel problem needed to be solved for data poisoning. The paper claims to craft better poisoning points using this methodology and shows improvements over previous methods that used various heuristics to approximate the solution to the bilevel problem. The authors show that poisoning points are transferable across various models and even hurt the performance of the models in the scenario where models are trained from scratch, which has been previously shown to be a difficult problem. The application and solution proposed by the method are very relevant to the community especially considering that data poisoning is emerging as a real threat to machine learning models. However, there are some problems with the contribution and the experiments as described below.

Strengths: The paper proposes to use a first-order meta-learning based scheme to solve the difficult optimization problem that appears in the data poisoning problem. The paper claims that solving this problem by jointly training network weights and poisons makes the poisoning points overfit a particular set of weights and their poisoning effect is reduced when tested against a different set of network weights (obtained through reinitialization). To eliminate this overfitting, the authors propose to use reinitialization and an ensemble of models that are trained using a certain schedule as described in Algorithm 1. Poisoning points generated through this procedure are shown to not only withstand re-training but also transfer to different models. The paper shows experiments for this on several large neural networks as well as public APIs. These could be used to highlight the importance of data poisoning in the machine learning community.

Weaknesses: Aside from experiments and demonstration of data poisoning on some real applications, the paper does not have much novelty. Several previous works have shown that data poisoning has a bilevel formulation and showed how to solve it for simple machine learning models. The first-order method described in the paper to approximately solve the bilevel problem has also been applied to the problem previously by [Munoz-Gonzalez et. al 2017] and original works go back to [Domke et. al 2012 and Maclaurin et. al 2015]. The paper uses a few step approximation and reverse mode automatic differentiation coupled with ensembling and reinitialization to solve the problem. As such the algorithm is not new however paper shows that the approach can be used to poison neural networks without handcrafted heuristics like water-marking which were essential in previous works. However, the authors provide no deeper reasoning for why their heuristics make all the difference. The authors report that unrolling a network for 2 steps works the best and doing larger unrolling steps actually hurts the success rate. This is counter-intuitive since reverse automatic differentiation based schemes should theoretically get better as the number of unrolling steps is increased. It would be useful if the authors could provide some explanation for this behavior. A similar explanation is essential for the number of models to consider in the ensemble during training. The paper is considering the problem of a clean-label attack which aims to affect the classification of a single target point. The experiment shows that to get a decent attack success at least 1% of the dataset must be poisoned. The reason so many points are needed is also not clearly explained by the authors. This is a crucial question in understanding the practical limits of such attacks and data poisoning in general. In the experiments, the authors only compare against feature collision attack and have not shown results for convex polytope attack which is also a popular clean label attack. It is important to see this result to be sure that their technique works for general data poisoning problems and not just a specific attack method. Some comparison against existing defense mechanism is also needed to be able to say that poison points learned from solving the bilevel problem are better than previously handcrafted heuristics. The authors briefly state that this attack can be robust to neighborhood conformity tests but have shown no experiments to justify the claim.

Correctness: The main algorithmic contribution of the work is to augment the reverse-mode automatic differentiation scheme with ensembling and a reinitialization schedule to solve the data poisoning problem. Their method seems to be working for neural networks with a large number of poisoning points required to hurt the classification of a single target point. This is better than the heuristics used by previous works which do not solve the bilevel problem well. However, the paper does not provide a deeper explanation of why their method works. This makes it difficult to understand whether the advantage really comes from solving the bilevel problem or using an ensemble model. The paper has only compared their approach to one of the attack methods and its not clear whether their approach will work for all data poisoning problems or not.

Clarity: The paper is clearly written and provides an explanation of their method in an easy to understand manner. However, the paper is missing the connection between their approach and reverse-mode automatic differentiation previously proposed in the literature. The paper also does not talk about other bilevel methods that exist in the literature which have better complexity than automatic differentiation based schemes. The paper sometimes makes some claims which are not clearly justified. For example, line 258 "may render", line 316 "robustness to data augmentation", some experiments are needed to fully justify these claims.

Relation to Prior Work: The paper discusses the previous works and shows the failure points of these. However, previous works on solving bilevel problems should also be discussed.

Reproducibility: Yes

Additional Feedback: Considering that the main contribution of the work is to augment the previously proposed methods based on automatic differentiation with ensembling and frequent reinitializations, it would be great if the authors could justify why their method works either theoretically or empirically. This could provide some insight into the reasons for the difficulty of data poisoning in deep neural networks. Some experiments on a different type of attack like convex-polytope are essential to say whether the advantage of the method is coming from the bi-level formulation or ensembling several models. Robustness to data sanitization defenses could be a better way to show that solving a bilevel problem to generate poisoning points is better than various heuristics. Some explanation about why the larger value of unrolling steps hurts the accuracy must be provided since its contrary to the theoretical claim that automatic differentiation approximates the solution better with larger unrolling steps. Clearly highlighting the connection of their method with previously proposed methods to solve bilevel problems is also important for readers unfamiliar with the works. I have read the authors' feedback on my questions, and my view of the paper has not changed much. There is the novelty of the proposed heuristic that allows poisoning of the neural networks trained from scratch. However, the impact of the heuristic is more practical than fundamental.


Review 2

Summary and Contributions: The paper proposes a clean label poisoning attack inspired by meta-learning to approximate bilevel optimization problem which is different from existing targeted data poisoning attacks based on feature collision. The significant advantage of using the scheme comes in “training from scratch scenarios” which also enables it to be effective in black-box settings. The proposed scheme can be used in other types of targeted poisoning tasks e.g., self-concealment and multiclass poison scheme which makes it a general approach applicable for attaining different adversarial objectives.

Strengths: The experimental evaluation evidences the benefits of the approach proposed by the authors with respect to other “clean-label” approaches for performing targeted poisoning attacks. As mentioned before, the attack proposed by the authors allows to cope with “training from scratch” scenarios, that were not considered in other clean-label targeted poisoning attacks. The authors also show experiments on transferability and include a section with other types of targeted poisoning attacks that are interesting.

Weaknesses: Although the experimental evaluation supports the benefits of the proposed approach to perform targeted poisoning attacks, the rationale and the description of the algorithm are unclear. From the explanations provided in Section 2.2 it is difficult to assess the soundness of the algorithm (see comments below). On the other side, the experimental evaluation just considers targeted poisoning attacks aiming to misclassify a single target sample in each attack. It is unclear how this algorithm would perform for indiscriminate attacks or attacks targeting a broader set of samples, similar to previous work in poisoning attacks relying on bilevel optimization.

Correctness: The description of the proposed algorithm in Section 2.2 does not provide enough information to assess its correctness. Although the empirical evaluation supports the usefulness of the algorithm it is unclear how the proposed algorithm relates to other techniques used to solve or approximate the solution of bilevel optimization problems.

Clarity: In general, the paper is well written except for Section 2.2.

Relation to Prior Work: The related work for data poisoning attacks is well covered in the paper. However, comparison with the literature on techniques to solve (or approximate the solution for) bilevel optimization problems can be improved.

Reproducibility: Yes

Additional Feedback: - As mentioned before, my main concern is about the rationale and the correctness of the algorithm. It is unclear that this algorithm is a good approach to approximate the solution of the bilevel optimization problem in equations (1) and (2). In this sense, previous works (including Munoz-Gonzalez et al.) rely on approaches that allow to approximate the solution of bilevel optimization problems providing, in some cases, some theoretical analyses and proofs of convergence to the optimal solution. See for example Pedregosa “Hyperparameter optimization with approximate gradient”, Franceschi et al. “Bilevel Programming for Hyperparameter Optimization and Meta-Learning”, Franceschi et al. Forward and Reverse Gradient-Based Hyperparameter Optimization”, Maclaurin et al. “Gradient-based hyperparameter optimization through reversible learning” or Domke “Generic methods for optimization-based modeling”. Although the authors have shown empirically that the algorithm is useful to perform targeted poisoning attacks, it is unclear that is a good solution to approximate the solution of bilevel optimization problems. - Related to the previous point, it would be interesting to compare the proposed approach to the one in Munoz-Gonzalez et al., even in the scope of linear classifiers or smaller neural networks. This could help to understand what are the benefits of the algorithm proposed by the authors compared to previous approaches. In this sense, it is unclear what are the benefits in terms of computational complexity of meta poison compared to the approximate algorithm proposed in Munoz-Gonzalez. - In the experimental evaluation, the performance of meta-poison compared to Shahafi et al.’s attack is clear, but it would be interesting to compare the computational cost of the two attacks. - Meta-poison works well for targeted attack scenarios, but given that previous attacks following a similar formulation (bilevel optimization) have performed indiscriminate and targeted attacks targeting a wider set of data points, it would be interesting to analyze the capabilities of meta-poison to perform those attacks. ------------------------------------------------------------- COMMENTS AFTER REBUTTAL: ------------------------------------------------------------- I have carefully read the rebuttal. I appreciate the effort put by the authors to clarify the reviewers’ comments. Some additional comments: - The experimental results on targeted attacks are convincing, although their scope is limited. I think that the results that the authors reported in the rebuttal on indiscriminate attacks are also very interesting. The authors said in the rebuttal that “practically, targeted attacks are more concerning”: I disagree with that. The kind of targeted attacks performed in the paper affect a very limited number of data points which, in many cases, may represent a minor threat. On the practicality of indiscriminate attacks, for example, in 2016, Tay, a chatbot developed by Microsoft suffered a (indiscriminate) poisoning attack that compromised the normal operation of the chatbot. If the paper is accepted, I strongly recommend the authors to include results on indiscriminate or targeted attacks targeting a wider set of examples. - Section 2.2 really needs to be improved. It looks like the heuristic proposed by the authors enables to craft poisoning points that generalize better compared to the case of solving the whole bilevel optimization problem. This is achieved by randomization and limiting the number of iterations in the inner problem to approximate the gradients in the outer optimization problem. But these aspects are not really covered in the current version of the paper. - In the rebuttal, the authors say: “Munoz-Gonzalez’s method differs from ours in that they unroll the whole training procedure…” This is untrue. Other papers like Biggio et al. or Mei and Zhu “Using Machine Teaching to IdentifyOptimal Training-Set Attacks on Machine Learners” rely on attack that unroll the whole training procedure. However, Munoz-Gonzalez et al. attack truncates the inner problem to a reduce number of iterations to approximate the gradient of the outer problem, similar to other approaches to approximate the solution of bilevel optimization problems as in Domke or MacLaurin et al. (see the references above). I have increased my score to 6 as I think that the paper has potential: the heuristic is interesting, although requires further motivation and justification. If the paper is accepted, I recommend the authors to use the reviewers’ comments to improve the paper.


Review 3

Summary and Contributions: This paper studies clean label targeted poisoning attacks where there is an adversary who adds some example to the training set with the goal of making the trained model to misclassify an specific instance. The adversary is restricted to use correct labels for the instances that it provides in the training set, hence the name clean-label attack. Previous attacks against neural networks only work for the settings where the initial weights of the neural network, which makes it to only work in scenarios that the initial weights are public, e.g. transfer learning. This paper relaxes this assumption and provides an attack that works against neural networks that use random initialization. The paper's shows a way to find poison points that are robust to the random initialization. They show that in some scenarios, their attack is successful with probability more than 50%, with poison budget 1%. The technique they use is quite simple. In order to deal with the randomness of initial weights, they train multiple networks on the training data with different initializations. The also optimize the poison points in parallel by taking the average of all the gradients coming from the multiple trained models. This way, the poison points are optimized for an ensemble of multiple models and have to work against many of them to be successful.

Strengths: -This is the first clean label targeted attack that works against neural networks that are trained from scratch. -The attack model is very realistic as they do not assume the knowledge of the initial weights of the network. They even show the attack could be applied to black-box training algorithms.

Weaknesses: -The scenarios that success of the attack is less than 50%, a simple ensemble method could be used to defend the attack. It seems that the success of attack in attacking the Google model is around 20% which could be circumvented by using multiple models. -The attack seems to be unstable when changing the architecture. For instance the attack on VGG does not succeed as much as the attack on other architectures. -On novelty of the paper: The ideas behind the attack seem to be simple and borrow ideas from the Metalearing literature. However, this is not necessary a bad thing as it shows simple ideas can be used to attack models. - The experiments of the paper are done only on neural networks and image classification tasks. It would be interesting to see the performance of attack on other architecture and classification tasks.

Correctness: The empirical method seems correct to me.

Clarity: The paper is written well.

Relation to Prior Work: The paper does a rather good job in comparing with previous work on clean-label poisoning. But I think they could also with other (not clean label) poisoning attacks as well.

Reproducibility: Yes

Additional Feedback: -In many scenarios the attack success is less than 50% (for example the case of black box Google model). Is it easy to prevent the attack by using an ensemble of models to prevent the attack? -I suggest to compare with other "dirty"-label attacks to see how efficient your attack works compare to them. It is not a fair comparison but still would be meaningful to see how many more poison samples you need to misclassify an instance. -Figure 7 is not clear. What is the 3d figure describing? -I think it would have been better if authors provide some discussion on the theoretical work on poisoning attacks. For example [1] studies clean label poisoning attacks with arbitrary targets and they show a general purpose attack. The attack is probably not very efficient for neural networks but I think it still worth the discussion. [1] http://proceedings.mlr.press/v98/mahloujifar19a/mahloujifar19a.pdf


Review 4

Summary and Contributions: This paper proposes a clean-label poisoning algorithm, MetaPoison, that slightly perturbs a small amount of training data so as to cause an unaltered target example to be misclassified. The proposed algorithm is shown to be effective for a variety of attacks on CIFAR10, while being quite robust to the choice of targeted model and training hyperparameters.

Strengths: - An effective an inconspicuous poisoning attack. Previous works had found clean-label poisoning to be quite hard to achieve, and this paper seems to bring some consistent improvements. - Thorough experiments demonstrating the effectiveness and robustness of the proposed algorithm - The paper is well-written, the different algorithm components are well-motivated and backed by experiments.

Weaknesses: - The algorithm assumes knowledge of the entire training dataset, which seems unrealistic. It would be nice to see how the attack performance degrades if the adversary only has partial knowledge of the training data. - The attack works for a fixed target image known at training time. While this may suffice for some applications (e.g., the watermarking idea at the end of the introduction), this seems insufficient for others (e.g., evading a facial recognition software, which will take a "live" picture of you at some point in the future).

Correctness: The experimental setup seems sound. I did not check the provided code.

Clarity: The paper is overall well-written and easy to follow. My only complaints here are: 1) The many text-wrapped figures are a bit of a pain for the reader. I understand that this comes from a need for space but it makes the paper hard to parse in somme cases. 2) The fact that the attacks only work for a single known target is an important limitation that should be acknowledged more clearly in the abstract and introduction. Currently, the abstract/intro could likely be misread as proposing a much stronger algorithm than the one presented. 3) Having poison budgets in percent on a log scale in Figure 4 is somehow very confusing to me.

Relation to Prior Work: The related work on this topic seems to be addressed appropriately, with one notable exception. The recent "Fawkes" paper (https://arxiv.org/abs/2002.08327, USENIX Security 2020) should be acknowledged as it operates in a very similar space, and seems to obtain comparable or better results in some cases (e.g., it seems that the attacks in Fawkes are more general, in that they do not apply to a single target image, nor do they assume that the adversary knows all the training data).

Reproducibility: Yes

Additional Feedback: - Why do you use the ReColorAdv perturbation? I haven't seen this perturbation function used before, so I wonder if there's any particular reason to use it here? - Step 16 of Algorithm 1 specifically mentions the use of Adam. The specific optimizer doesn't seem to be a core part of your algorithm so I would suggest remove this specificity from the algorithm description.


Review 5

Summary and Contributions: This paper presents a new solution to create clean-label poisoned data to fool a classifier for certain input samples. Compared to past works, this solution unrolls the training pipeline during the optimization of poisoned data generation. An ensemble of models are created from different training epochs, and network re-initialization are used for data creation.

Strengths: + Higher success rates compared to past works. + Practical evaluation in Google Cloud ML services

Weaknesses: - Technical novelty is relatively limited - Descriptions about the methods are not very clear - Some arguments about the contributions are too strong.

Correctness: Seems to be

Clarity: Can be improved, especially for the methodology part

Relation to Prior Work: Yes

Reproducibility: Yes

Additional Feedback: This paper presents an interesting attack technique for clean-label data poisoning. The comparisons with prior works are clearly described, and empirically verified. I particularly like the attacks against Google Cloud AutoML. Below are some improvements the authors can consider: 1. Some contributions claimed by the authors (Lines 72 - 83) are too strong. For instance, in the third one, the authors mentioned this is the first work to poison neural networks training from scratch. This is clearly incorrect. A lot of works have been proposed for this goal, although they are not clean-labels. The authors should add such condition (clean-label) when claiming the "first". Another example is the fourth one, blackbox transferable attacks. Such setting has been considered in past works (e.g., Zhu et al. 2019). This is not new. I do not know why this is also regarded as a contribution by the authors. 2. Technically the novelty is limited. The adversarial loss function is from past work, the training pipeline unrolling is from past work, the perturbation generation is also based on existing technique. The novelty here lines in the adoption of model ensembling and network re-initialization, which is minor. 3. The methodology description part is a bit difficult to understand. For instance, in line 120: "a computation graph that explicitly unrolls 10^5 SGD steps...". This is confusing for readers who do not have deep understanding about the clean-label attacks. What does the computation graph refer to? How is unrolling used to achieve this goal? Without such introduction, it is hard to get the key point and innovations of this approach. 4. I am a bit worried about the computation cost, as the approach requires to train multiple models. I am not sure if it needs to train those models separately, or use train it once and collect the breakpoint at different epochs? This is not stated in the paper. If it is the first case, then the performance cost will be large especially when the model is complex. Line 189 mentioned the cost is 6 GPU-hours. Does it include the model training? I am also interested to see the cost for Imagenet. 5. For comparisons with past works, this paper only considered [Shafahi et al, 2018]. What about others like [Zhu et al. 2019]? This is a newer attack. For transferability, there are no comparisons with [Zhu et al, 2019], which is also focusing on transferability.

[Author Response · NeurIPS 2020]

Reviewers, thanks for your feedback. We want to reiterate that our main contribution is *practical* poisoning, demon-
strated by being first to poison DNNs trained from scratch & first to break an industrial API (Google) via poisoning.
*R1 & R2*: **Justification for two-step approximation** Our decision to unroll the inner objective for a few steps (K=2)
vs. the whole training ($K \approx 10^5$) is a key distinction between our approach and that of Domke, MacLaurin, & Munoz-
Gonzalez. Our ablation study vs. K in Supplementary Material Fig 7 confirms that gains diminish beyond K=2,
corroborating with prior work on higher-order backprop, e.g. Finn et al MAML. There's two reasons for this. First,
Shaban et al "Truncated Back-propagation for Bilevel Optimization" §3.1 theoretically show the approximation error
of few-step gradient evaluations to decrease exponentially with K, so the gradient is already well approximated with
a few steps. Meanwhile, unrolling many steps leads to numerical instability due to the ill-posedness of the gradient
operator, as observed in MacLaurin et al even for convex problems. Early stopping of the approximation prevents
this instability. Second, Franceschi et al "Bilevel Programming for Hyperparameter Optimization and Meta-Learning"
§5.1 shows that computing the exact bilevel solution (our Eqs. 1-2) can lead to overfitting the outer objective. They
show instead that the approximate gradients from small K act as an implicit regularizer for *generalization*, which is
more desirable here than the exact solution b/c the initialization, SGD order, architecture, and other nuisance variables
differ when the bilevel objective is evaluated by the victim. Lastly, small K is cheap, while $K \approx 10^5$ is intractable.
In summary, the best solution to *poisoning problem* is one that generalizes and computes fast; both point to a small
K approximation. **Justification for reinitialization and ensembling** As R1 correctly points out, reinitialization and
ensembling are regularization techniques to help poisons further build invariances to nuisance variables above by seeing
more surrogate models, yielding better generalization. *We've performed an ablation study (which will be added to the*
*revision) showing that neither reinitialization w/o ensembling, nor emsembling w/ fixed initialization, suffice to yield*
*poisoning success.* Algorithmic details will be more clearly hashed out in §2.2. **Related bilevel work** We will relay the
discussions above, highlight relation to other bilevel methods, and include the valuable citations from R2.
*R1*: **Why more unrolls hurts** Actually, Supplementary Material §G says $K \geq 2$ "seems to not affect the result much."
**Why 24-model ensemble** Our ablation on ensemble size in Supplementary Material §E sees gains diminsh beyond
24. **1% budget needed for decent success** We need only 0.04% (20 poison) budget to achieve ∼90% success for
fine-tuning (Fig 3) and ∼20% success for training-from-scratch (Fig 4)—alarming levels for industrial security. **Com-**
**pare to convex polytope (CP)** *Using Zhu et al's setup for CP, MetaPoison gets 60% success on ResNet20 transfer*
*learning*, compared to Zhu's 52%. **Defense evaluation** We obtained code for Peri et al's "Deep $k$NN Defense Against
Clean-label Poisoning Attacks" (which reports 100% detection of CP attacks w/ minimal false positives), and evaluated
on MetaPoison. *Deep-kNN fails to detect any MetaPoisons at any k*. This makes sense as Fig 3 shows MetaPoison's
features don't lie in the target's neighborhood, whereas FC & CP's do. These and the CP results above will be added.
**Line 316**'s substantiated by Fig 5. Our method is also **reverse-mode autodiff** based like the literature. We're unaware
of **better complexity non-autodiff** methods applicable to DNNs and are open to suggestions.
*R2*: **Indiscriminate and multi-target attacks** Per your request, *we ran new attacks along the indiscriminate/multi-*
*target/single-target spectrum, including 1. fully indiscriminate (error-generic), 2. indiscriminate for specific class*
*(error-specific), 3. multiple (5) distinct target objects, 4. multiple (>10) augmentations of same target object.* Briefly,
MetaPoison did okay on the more indiscriminate attacks (error-increase of 8% on attack 1 and 15% on attack 2 using 5%
budget) and quite well on the more targeted attacks (success of 34% on attack 3 and 52% on attack 4 using 1% budget).
Practically, targeted attacks are far more concerning, as indiscriminate attacks can be easily detected by evaluating on a
holdout set. Detailed plots will be added to the revision. **Compare to Munoz-Gonzalez** Munoz-Gonzalez focuses on
proof-of-concept using a toy setup (fixed initialization, small dataset of 1000, unbounded perturbation, label flips) rather
than practicality, whereas MetaPoison focuses on practicality by considering real-world constraints. Munoz-Gonzalez's
method differs from ours in that they unroll the whole training procedure, don't do reinitialization or ensembling, and
use nonstochastic GD. *Our replication of their method and setup yielded comparable performance (6% indiscriminate*
*error above label-flipping with 5% poisons) on an (unrealistic) victim with the same initialization but null performance*
*(0%) on victims with different initializations.* Complexity-wise, Munoz-Gonzalez reports O(K) (K = num training steps)
per outer step per poison while ours is O(1) since we fix K=2 and ensemble=24. **Computation cost vs. Shafahi** Per
poison, MetaPoison takes 2 (unrolling steps) x 2 (backprop thru unrolled steps) x 60 (outer steps) x 24 (ensemble size)
= 5760 forward+backward propagations. In contrast Shafahi reports 12000 forward+backward props. Thus MetaPoison
has similar cost if we discount training of the surrogate models. This is reasonable given that pretrained surrogate
models along with intermediate checkpoints can be precomputed and reused. Our latest code samples weights from a
database of checkpoints w/ no degradation.
*R4*: **Ensembling defense** Recall that success is averaged over multiple runs *and* target images. For a *specific* target
image, success tends to be quite stable; e.g., success for bird 0 and 6 is 100% over 4 runs on Google AutoML (each
w/ different architectures). This suggests that ensembling will not be effective. **Unstable across architectures** It's
true poisons don't transfer to all models equally, though transferabilty to VGG is quite good–87% from ConvNetBN.
**Non-image tasks & dirty-label attacks** While outside this paper's scope, these are important and will be a focus of
future work. **Fig 7 (right) unclear** We'll clarify this figure more in the revision. **Theoretical poisoning** We'll expand
our related works with this and other theoretical works.

[Meta-Review · NeurIPS 2020]

I want to thank the authors for preparing the rebuttal and for sharing their concerns about one of the reviews. This paper was heavily discussed among all the reviewers during the post-rebuttal discussion phase. Given the paper's borderline scores, we also requested additional emergency reviews for this paper -- I hope that the authors will find this additional feedback useful. During the discussion phase, all the reviewers did acknowledge the importance of the proposed practical and scalable method for poisoning the neural networks. Based on the post-rebuttal discussions, some of the reviewers have updated their reviews, providing additional comments. I would like to strongly encourage the authors to incorporate the reviewers' feedback when preparing the paper's final revision.